# Mirror Mirror on the Wall, Have I Forgotten it All?
# A New Framework for Evaluating Machine Unlearning

## Abstract

Machine unlearning methods take a model trained on a dataset $\mathcal{D}$ and a forget set $\mathcal{D}_f$ then attempt to produce a model as if it had only been trained on $\mathcal{D} \setminus \mathcal{D}_f$. We empirically show that an adversary is able to distinguish between a mirror model (a control model produced by retraining without the data to forget) and a model produced by an unlearning method across representative unlearning methods (Foster et al., 2023; Graves et al., 2020; Chundawat et al., 2023; Zhang et al., 2024). We build distinguishing algorithms based on evaluation scores in the literature. We contribute a strong formal definition for machine unlearning called *computational unlearning*. Computational unlearning is defined as the inability for an adversary to distinguish between a mirror model and a model produced by an unlearning method. Our computational unlearning definition allows us to prove feasibility results and demonstrate that current methodology in the literature —such as differential privacy — fundamentally falls short of achieving computational unlearning. We leave achieving practical computational unlearning for future work.

## 1 Introduction

Machine learning models require massive amounts of training data. Data is collected by scraping publicly available web content (David, 2023; Mehrotra & Couts, 2024; Weatherbed, 2024), purchasing access to private databases (Knibbs, 2024; OpenAI, 2024b; 2023b; David, 2024; Springer, 2023; OpenAI, 2023a; Atlantic, 2024; OpenAI, 2024a), and collecting data on their own to assemble training datasets (Schuhmann et al., 2022; Touvron et al., 2023; Brown et al., 2020). Due to the massive scale, datasets cannot be thoroughly vetted and may contain data that is copyrighted, inaccurate, protected, or contain otherwise undesirable information.

Legal protections exist for those who wish to protect their privacy, copyrighted content, and financial history in multiple countries. Examples include the EU GDPR (right to be forgotten) (European Parliament & Council of the European Union), US DMCA (copyright infringement takedown) (United States Congress, 1996), US FCRA (corrections to credit history) (United States Congress, 1970), and US HIPAA (corrections to personal health data) (Centers for Medicare and Medicaid Services, 1996). Specific instances of training data may also be illegal on their own: for example, it is illegal to possess child sexual abuse material (CSAM) in the US and in many other jurisdictions. Despite this, popular datasets (Schuhmann et al., 2022) used to train models like Stable Diffusion contained illegal CSAM (Thiel, 2023). Further, prior work has established the threat of data poisoning attacks that create backdoors in models (Goldwasser et al., 2022; Gu et al., 2019; Li et al., 2022). This means that model data may be intentionally corrupted by an adversary.

These threats can be addressed by re-training the model from scratch without the offending data. However, since training large models is capitally and computationally intensive, a major area of interest is *machine unlearning:* efficiently removing traces of the offending data, known as the as the *forget set*, without training a new model from scratch (a *control* model) (Cao & Yang, 2015; Abadi et al., 2016; Golatkar et al., 2019; Bourtoule et al., 2020; Graves et al., 2020; Gupta et al., 2021; Ullah et al., 2021; Nguyen et al., 2022; Chundawat et al., 2023; Foster et al., 2023). However, as other

authors have noted, there is still work to be done on rigorous evaluations of unlearning (Hayes et al., 2024).

## 1.1 OUR CONTRIBUTIONS

This work consists of three major contributions: (1) a new formal adversary specification and framework for evaluating unlearning, (2) empirical evaluation of inexact unlearning methods and (3) several feasibility results on achieving computational unlearning. We leave achieving practical computational unlearning to future work.

**Computational unlearning framework.** Though unlearning is understood to be the process of removing information learned from specific data points, there is need for rigorous adversarial definitions to evaluate unlearning. Our primary contribution is a new formal definition and framework for evaluating unlearning called *computational unlearning* that we detail in §3. In brief, computational unlearning tests the ability of an adversary to distinguish between a model produced by an unlearning method (an *unlearned* model) and a model trained from scratch with the forget set removed (a *control* or *mirror* model). If the adversary is only able to do so with negligible probability, then we say that the unlearning method achieves computational unlearning. Because the adversary is unable to distinguish between the control and unlearned models, it follows that all information about the forget set has been "deleted" by the unlearning method. The game is defined in both a white-box (i.e. adversary has full access to model parameters) and a black-box (i.e. adversary only has API access to model) setting. This is distinct from and complementary to the the definition posed by Hayes et al. (Hayes et al., 2024), which is focused on privacy leakage and distinguishing between the forget set and the set of data the model has never seen. We pose a stronger adversary that controls the original training set and selects the forget set.

**Many unlearning methods do not achieve indistinguishability.** We construct two scoring methods `MIAScore` and `KLDScore` in §4.1 which an adversary can use to distinguish between an unlearned model and a model that has never seen the forget set. We study previously proposed unlearning methods (Foster et al., 2023; Graves et al., 2020; Chundawat et al., 2023; Zhang et al., 2024) and show that each fail to achieve computational unlearning for ResNet-18 models (He et al., 2015b) trained on CIFAR-10 (Krizhevsky, 2009) in §4. We also experiment with how distinguishing rates are affected by the forget set size and unlearning method hyperparameters.

**Theoretical implications of computational unlearning.** We describe several implications of our computational unlearning framework in §5. We first show that any deterministic computational unlearning algorithms must achieve *perfect unlearning* (i.e. it must produce the exact same model as retraining) and discuss implications for heuristic and certified removal unlearning methods. Second, we show that using differential privacy to achieve black-box computational unlearning leads to utility collapse (i.e. utility must be equivalent to a model that is randomly initialized).

## 2 MOTIVATION

### 2.1 ADDRESSING OVERFORGETTING AND UNDERFORGETTING

Many machine unlearning works attempt to justify their approach by optimizing some *unlearning score*. Membership inference attack (MIA) scores, formalized by Shokri et al. (Shokri et al., 2017), are a common way to evaluate the performance of machine unlearning algorithms in literature and attempt to predict if a model was trained with a given data point. Applying MIA scores to evaluate unlearning makes intuitive sense: if a model has unlearned data it should have a low MIA scores, similar to a model that never saw the data. As a result, many heuristic machine unlearning proposals are specifically designed to minimize these MIA scores (Graves et al., 2020; Chundawat et al., 2023; Foster et al., 2023).

Framing machine unlearning in a score-based manner is attractive: it provides an easy way to facilitate comparison, and it also satisfies intuitive beliefs about how the model should behave after unlearning. However, score-based definitions do not address consequences arising from discrepancies in knowledge between an unlearned model and a control model. These discrepancies can be categorized as

*overforgetting* and *underforgetting*. Essentially, overforgetting results in losing too much information, while underforgetting results in retaining too much information. Unlearning methods that are prone to overforgetting produce models that perform worse on the retained information than a control model, while unlearning methods prone to underforgetting produce models that perform better on the retained information than a control model.

The consequences of this issue can be seen when applying unlearning to backdoor attacks (Goldwasser et al., 2022; Li et al., 2022). Unlearning is a possible defense as a good unlearning method should remove all knowledge of the backdoor from a model. However, prior work has shown that existing unlearning methods fail to actually remove a backdoor from a model (Pawelczyk et al., 2025). In other words, these unlearning methods are prone to *underforgetting* and thus can not be trusted to fully remove a backdoor from a model.

### 2.2 MACHINE UNLEARNING IS INDISTINGUISHABILITY

We claim that unlearning needs a new definition that accounts for the aforementioned issues. An unlearning method should produce a model that is *indistinguishable* from a control model. This indistinguishability implies that the unlearned model has not overforgotten or underforgotten.

Additionally, how indistinguishability is measured should be meaningful — better unlearning methods will produce unlearned models that are *harder* to distinguish from a control model. This idea is not new and features in prior work (Zhang et al., 2024; Guo et al., 2023; Foster et al., 2023; Hayes et al., 2024) but is not measured directly. We propose doing so here. In other words, no efficient (p.p.t., or probabilistic polynomial time) adversary $\mathcal{A}$ should be able to distinguish between a model produced by an unlearning method and a model trained without the forget set.

**Motivation with $k$-NN.**   Observe that our desired functionality is readily apparent in the $k$-nearest neighbors ($k$-NN) algorithm (Fix & Hodges, 1989). Since $k$-NN requires memorizing all training data it immediately admits an unlearning algorithm: simply delete the training examples you wish to forget. This produces a model that is indistinguishable from a control.

## 3 FORMALIZING UNLEARNING

We propose *computational machine unlearning* as a formal way to capture that machine unlearning is indistinguishability. Unlike prior machine unlearning scores, our definition is defined as a security game, inspired by the cryptographic notion of semantic security and indistinguishability under chosen plaintext attack (IND-CPA) (Boneh & Shoup, 2023). Instead of considering an MIA score, computational unlearning considers the ability of an adversary to distinguish between an unlearned model and a control model.

### 3.1 PRELIMINARIES

Let $\mathcal{U}$ be the universe of all possible data, and $d \in \mathcal{U}$ be a particular data point. Let $\mathcal{D} \subseteq \mathcal{U}$ be our entire training dataset with $\mathcal{D}_f \subseteq \mathcal{D}$ be the forget set. Let $\mathcal{H}$ be our hypothesis space of possible models, with $h \in \mathcal{H}$ being a particular model.

**Definition 1** (Learning scheme)**.** We formally define a *learning scheme* as a tuple of probabilistic polynomial time (p.p.t.) algorithms (init, learn, infer):

- init($1^\lambda$) $\rightarrow h$: randomly samples some initial model $h$. The notation $1^\lambda$ simply denotes that there are $\lambda$ copies of the symbol 1 written on the input tape of the Turing machine and 0 in every other location. This ensures that init runs in polynomial time with respect to $\lambda$, a cryptographic formality.

- learn($h, \mathcal{D}$) $\rightarrow h$: given some initial model $h$, performs some model update process with respect to the training set $\mathcal{D}$.

- infer($h, d$) $\rightarrow \mathbb{R}^n$: performs some inference procedure with the given model $h$ on the provided data point $d$.

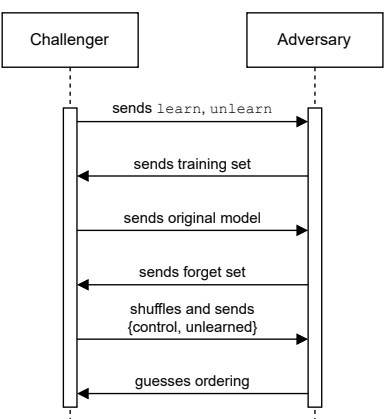

Figure 1: Overview of the security game for computational unlearning.

**Definition 2** (Forgetting learning scheme). We likewise define a *forgetting learning scheme* as a tuple of p.p.t algorithms (init, learn, infer, unlearn) such that it is a learning scheme with an additional unlearn($h, \mathcal{D}_f$) → $h$ algorithm.

**Definition 3** (Negligible function). We define negl($\lambda$) to be a function that is *negligible* in terms of a security parameter $\lambda$. We borrow the definition of a negligible function from cryptography — namely, that a function $f : \mathbb{Z}_{\geq 1} \to \mathbb{R}$ is negligible if and only if for all $c > 0$ we have $\lim_{n \to \infty} f(n)n^c = 0$.

### 3.2 COMPUTATIONAL UNLEARNING

We now formally define *computational unlearning* in both white-box and black-box settings.

**Definition 4** (White-Box Computational Unlearning). We consider the following experiment:

1. $\mathcal{C}$ sends the description of the forgetting learning scheme (i.e. the learn and unlearn algorithms).
2. $\mathcal{A}$ chooses $\mathcal{D}$ and sends it to $\mathcal{C}$.
3. $\mathcal{C}$ computes $M_o \leftarrow$ learn(init($1^\lambda$), $\mathcal{D}$) and sends ($M_o$, learn, unlearn, $\mathcal{D}$) to $\mathcal{A}$.
4. $\mathcal{A}$ selects a forget set $\mathcal{D}_f \subset \mathcal{D}$ and sends $\mathcal{D}_f$ to $\mathcal{C}$.
5. $\mathcal{C}$ computes $M_u \leftarrow$ unlearn($M_o, \mathcal{D}_f$) and computes $M_c \leftarrow$ learn(init($1^\lambda$), $\mathcal{D} \setminus \mathcal{D}_f$).
6. $\mathcal{C}$ samples a random bit $b \xleftarrow{\$} \{0, 1\}$. If $b = 0$, $\mathcal{C}$ sends $[M_c, M_u]$. If $b = 1$, $\mathcal{C}$ sends $[M_u, M_c]$.
7. $\mathcal{A}$ computes a guess $b'$ and sends $b'$ to $\mathcal{C}$. $\mathcal{A}$ wins the game if $b' = b$.

We say that an unlearning algorithm is a *white-box computational machine unlearning algorithm* if

$$\mathbb{P}\left(b' = b\right) < \frac{1}{2} + \text{negl}(\lambda)$$

We denote this computational indistinguishability by saying $M_u \overset{c}{\approx} M_c$. This game is illustrated in Figure 1.

**Definition 5** (Black-Box Computational Unlearning). We consider the white-box computational unlearning experiment from Definition 4, modifying item 6 as follows: $\mathcal{C}$ samples a random bit $b \xleftarrow{\$} \{0, 1\}$. If $b = 0$, $\mathcal{C}$ sends $[O_{M_c}, O_{M_u}]$ where $O$ is an oracle that allows $\mathcal{A}$ to call infer on the underlying model. If $b = 1$, $\mathcal{C}$ sends $[O_{M_u}, O_{M_c}]$.

As with Definition 4, we say that an unlearning algorithm is a *black-box computational machine unlearning algorithm* if we have $\mathbb{P}(b' = b) < \frac{1}{2} + \mathrm{negl}(\lambda)$.

*Remark* 6 (Threat Model). This definition intuitively captures a setting inspired by the GDPR process: we assuming the adversary is a user who can select which data should be deleted (i.e. the set of items to be deleted is adversarially-controlled) as in (Hu et al., 2024). We also acknowledge that this game defines a very strong adversary and that a real-world adversary may not have access to the full training set, the description of the unlearning algorithm, or other information provided in this game. However, each of these alternatives envisions a strictly *weaker* adversary than our computational learning game, meaning that an unlearning method that achieves computational unlearning would still be indistinguishable from a control model in these scenarios.

## 4    EMPIRICAL RESULTS

We now present empirical distinguishers for $\mathcal{A}$ to evaluate if unlearning methods from literature achieve computational unlearning. We experimentally demonstrate the effectiveness of these distinguishing algorithms on heuristic unlearning and certified removal methods.

### 4.1    DISTINGUISHER SCORES

Each distinguisher for $\mathcal{A}$ uses a *scoring function* to separate $M_c$ from $M_u$. The scoring function takes in the original model $M_o$, a candidate model $M \in \{M_1, M_2\}$, the training set $\mathcal{D}$, and the forget set $\mathcal{D}_f$. The scoring function then outputs a value $s$ that is used to determine if the candidate model is $M_u$ or $M_c$.

**Scoring with membership inference attacks.**    As described in §2.1, membership inference attacks (MIA) are a common method for evaluating the performance of a given unlearning algorithm and several unlearning methods are justified by reducing them as much as possible. However, we are able to leverage these scores to distinguish an unlearned model from a control model *because the unlearning method will often produce models whose MIA scores are out of distribution*. We propose that an unlearning algorithm should achieve similar MIA scores to a model that never saw the forget set rather than attempting to absolutely minimize it. In experiments, we use the approach of Shokri et al. (Shokri et al., 2017) for computing MIA scores using the same implementation as Foster et al. (Foster et al., 2023). We refer to this scoring algorithm as `MIAScore`.

**Scoring with Kullback-Leibler divergence.**    We also present a novel scoring method `KLDScore`. We drew inspiration from the fact that Certified Removal bounds the KL-Divergence between different models. To calculate the score, $\mathcal{A}$ calculates the KL-Divergence between the inferences of the original model $M_o$ and the candidate model $M$ (such as on instances in or near the forget set). This provides a measure of how different the behaviors of $M$ and $M_o$ are. In practice, we find that models produced by unlearning methods have much lower divergence from the original model than a control.

$$\mathtt{KLDScore}(M_o, M, \mathcal{D}, \mathcal{D}_f) = \sum_{x_i \in \mathcal{D}_f} D_{\mathrm{KL}}(M(x_i + \mathcal{N}(0, 0.1)) \parallel M_o(x_i + \mathcal{N}(0, 0.1))) \tag{1}$$

where $\mathcal{N}(0, 0.1)$ represents Gaussian noise with mean 0 and variance 0.1.

**Choice of decision rule.**    $\mathcal{A}$ will compute $b'$ using the results from one of the aforementioned scoring algorithms. By Definitions 4 and 5, $\mathcal{A}$ is free to use prior knowledge of `learn`, `unlearn`, $\mathcal{D}$, and $\mathcal{D}_f$ in the decision rule.[1]

### 4.2    EXPERIMENTAL RESULTS

We evaluate the distinguishers via their success rates in differentiating between $M_u$ and $M_c$. For this, we present our findings from two experiments: one varying the size of the forget set $\mathcal{D}_f$ and the other varying the $\sigma$ parameter from Certified Deep Unlearning. We also show that unlearned models

---

[1]See Kerckhoffs's principle in cryptography.

which obtain closer scores to a control model are less prone to underforgetting. All experiments were run using an Intel Xeon Gold 6330 and a NVIDIA A40. All results are statistically significant (i.e. a 95% confidence interval under a Beta distribution with the Jeffries prior does not contain 50%). Implementation details for these experiments can be found in Appendix B.

**Forget set size.**    We evaluated the effect of the forget set size on four different unlearning techniques. We used three heuristic methods and the approximate technique Certified Deep Unlearning (CDU), all discussed in Appendix A. For each method, a random subset of $\mathcal{D}$ was chosen as the forget set. We varied the forget set size to evaluate its effect on the ability of $\mathcal{A}$ to distinguish between $M_u$ and $M_c$ and correctly guess $b'$ using the distinguishing algorithms discussed above. We ran 128 trials, each with a different randomly selected forget set. We found that with increased forget set size the adversary was able to correctly guess $b'$ with higher frequency, but always maintained above a 60% success rate at every forget set size. As we hypothesized, many heuristic unlearning techniques over-minimized `MIAScore` during their process of unlearning: for all heuristic unlearning methods the decision rule assigns a lower `MIAScore` score to $M_u$ (except for SSD (Foster et al., 2023) with greater than 30 forget set examples).

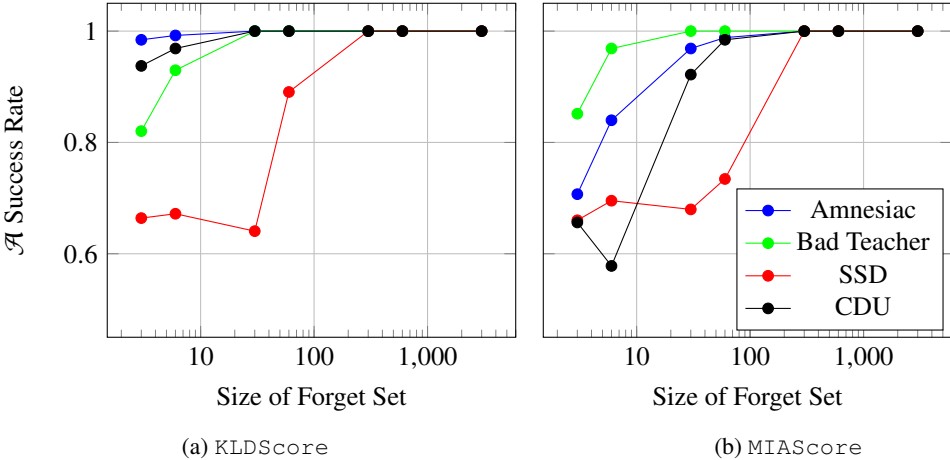

(a) `KLDScore`                   (b) `MIAScore`

Figure 2: Forget set size against adversary success rate using `KLDScore` and `MIAScore` distinguishers.

We also explored *classwise* unlearning, where an entire class in $\mathcal{D}$ is chosen as the forget set. We found it always possible to distinguish in this setting (i.e. 100% adversary success rate under both distinguishers). This is unsurprising given our results on the impact of forget set size. Recall that CIFAR-10 has 50,000 images in the training set, distributed evenly across 10 classes; forgetting an entire class amounts to a forget set size of 5,000 (Krizhevsky, 2009).

**Dependence on $\sigma$.**    We additionally explored the relationship between computational unlearning and certified removal's privacy parameters. For this we examined $\mathcal{A}$'s `KLDScore` for certified deep unlearning (CDU) from Zhang et al. (Zhang et al., 2024) with different hyperparameters. The CDU method is based on a single hyperparameter $\sigma$, derived from $\epsilon$ and $\delta$ values, that represents the magnitude of noise used. We follow the hyperparameters from the CDU published experiments (Zhang et al., 2024), including a random forget set of 1000 data points. We then varied $\sigma$ from $10^{-5}$ to $10^{-1}$ in powers of 10 running 128 trials at each value.

In our experiments we found that the adversary was able to distinguish using `KLDScore` with 100% accuracy for all choices of $\sigma$. We found as $\sigma$ increases the unlearned model's `KLDScore` also increases (see Figure 3). Since the control model has no dependency on $\sigma$, an adversary can distinguish with extremely high success rate by choosing a decision rule appropriate for the chosen value of $\sigma$. This relationship does imply there is a point of intersection (between 0.001 and 0.01) where the `KLDScore` score for $M_u$ and $M_c$ should be very close, making it harder to distinguish using `KLDScore`. We believe understanding the intersection constitutes an interesting topic for future work.

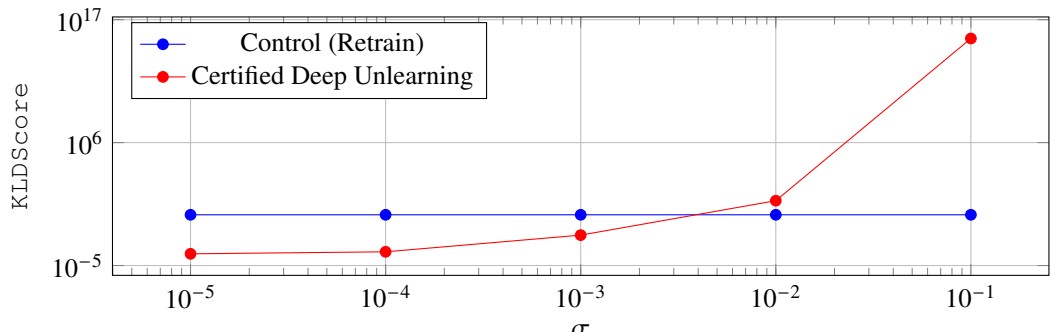

Figure 3: Certified Deep Removal against `KLDScore` for different values of $\sigma$.

**Closer scores are better.** While perfect indistinguishability remains out of reach, progress can be made by producing unlearned models that are *closer* to a control. We show that unlearning methods which achieve distinguisher scores closer to that of the control model are less prone to the negative consequences described in §2.1. We use the BadNets attack (Gu et al., 2019) with a fixed poison rate of 10% on CIFAR-10 (Krizhevsky, 2009). We tested the unlearning methods specified in Appendix A, comparing the closeness of $M_u, M_c$ to the accuracy of $M_u$ on poisoned data. This closeness was measured via the absolute value of the difference between the `KLDScore` scores of $M_u$ and $M_c$. As we see in Figure 4, larger deviations are directly correlated with higher performance on poisoned data.

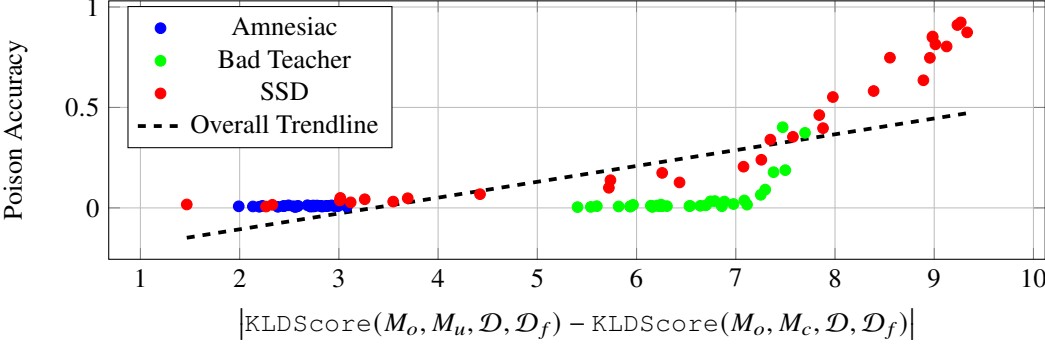

Figure 4: Absolute difference in `KLDScore` against accuracy of data on poisoned samples. The trendline (black) was computed with ordinary least-squares and has a $t$-value of 9.864, indicating that there is over a 95% probability that unlearned models with closer `KLDScore` scores to the control have better forgotten the backdoor.

## 5 THEORETICAL ANALYSIS

We now show several interesting consequences of our computational unlearning definition. or proofs of all the following theorems and corollaries, see Appendix C.

We begin by showing that $k$-NN admits a white-box computational unlearning algorithm in line with the technical intuition from §2.2.

**Theorem 7** ($k$-NN admits white-box computational unlearning). *There is an efficient white-box computational unlearning algorithm for $k$-NN models.*

We first show that for entropic machine learning algorithms (e.g. stochastic gradient descent) there are no deterministic algorithms that can achieve computational unlearning. This result means that many heuristic unlearning methods can *never* admit computational unlearning algorithms. Secondly, we show that differentially private algorithms can achieve computational unlearning at the cost of collapsing model utility.

## 5.1 DETERMINISTIC COMPUTATIONAL UNLEARNING DOES NOT EXIST

We now show that a forgetting learning scheme that is entropic must have a randomized unlearning algorithm. Additionally, we show that a forgetting learning scheme that is deterministic must achieve perfect unlearning. Because forgetting learning schemes that are entropic must must be randomized and because forgetting learning schemes that are deterministic must be perfect, we say that *deterministic computational learning does not exist*.

Before beginning, we define *entropic learning schemes* and *perfect unlearning*.

**Definition 8** (Deterministic learning scheme). A learning scheme is deterministic if the distribution of models produced by $\texttt{learn}\left(\texttt{init}\left(1^\lambda\right), \mathcal{D}\right)$ has Shannon entropy of 0.

**Definition 9** (Entropic learning scheme). A learning scheme is *h-entropic* if the distribution of models produced by $\texttt{learn}\left(\texttt{init}\left(1^\lambda\right), \mathcal{D}\right)$ has Shannon entropy greater than or equal to $h$. In the absence of a particular value specified for $h$, we take $h$ to be 1 bit.

*Remark* 10. If a learning scheme is entropic, it cannot be deterministic. For all practical purposes, learning schemes are either deterministic (i.e. $k$-nearest neighbors) or entropic (i.e. randomly initialized neural nets trained under stochastic gradient descent).

**Definition 11** (Perfect unlearning). We say a forgetting learning scheme achieves *perfect unlearning* algorithm if, for all $M = \texttt{learn}(\texttt{init}(1^\lambda), \mathcal{D})$, the following always holds:

$$\texttt{learn}(\texttt{init}(1^\lambda), \mathcal{D} \setminus \mathcal{D}_f) = \texttt{unlearn}(M, \mathcal{D}_f)$$

This is to say, $\texttt{unlearn}$ is perfect if it produces *exactly the same* model as retraining on the retain set.

Recall that in our definition, $\mathcal{A}$ is given the description of the unlearning method ($\texttt{unlearn}$) and also has access to the original model $M_o$. Intuitively, this means that an adversary can simply *run the unlearning method on its own*.

Because the unlearning algorithm is deterministic and the learning scheme is entropic, this means that only one of the two models will exactly match the adversary's own computed result with high probability and allow the adversary distinguish with non-negligible probability.

**Theorem 12.** *There are no deterministic computational unlearning algorithms for entropic learning schemes.*

We now show that a forgetting learning scheme that is deterministic and achieves computational unlearning must be perfect. The intuition for this result is similar: the adversary has access to $\texttt{learn}$, the description of the learning algorithm, and has access to the $\mathcal{D} \setminus \mathcal{D}_f$. This means that the adversary can compute the control model on their own, use its own control model to identify the control model provided by the challenger, and distinguish with non-negligible probability.

**Theorem 13.** *Let $\mathcal{L}$ be a forgetting learning scheme that is deterministic. Then if it satisfies the computational unlearning notion of Definitions 4 and 5 it must perfectly unlearn under Definition 11.*

*Remark* 14 (Viability of computational unlearning methods). These results constrain the space of learning algorithms that are compatible with unlearning. To reiterate: Theorem 12 shows that entropic learning schemes that are forgetting and achieve computational unlearning must have a randomized unlearning method. In the opposite direction, no deterministic learning algorithms can support entropic unlearning algorithms. Any deterministic learning scheme that is forgetting and achieves computational unlearning must implicitly realize a *perfect* unlearning scheme, as noted in Theorem 13. As a consequence of these findings, any forgetting learning schemes that achieves computational unlearning must either be perfect, or both the learning and unlearning process must inherently be randomized. Note that Certified Deep Unlearning (Zhang et al., 2024) and many heuristic unlearning methods we studied in §4 are not randomized and are not perfect. Thus, they can never achieve computational unlearning.

## 5.2 COMPUTATIONAL UNLEARNING FROM DIFFERENTIAL PRIVACY COLLAPSES UTILITY

One natural approach to constructing computational unlearning uses techniques from differential privacy (Dwork & Roth, 2014).

While differentially private learning algorithms imply the existence of black-box computational unlearning, the parameters choices required to achieve computational unlearning will lead to utility collapse for the resulting models. We show that the $\epsilon$ and $\delta$ parameters must be phrased in terms of $\lambda$ and that values needed to obtain security imply unacceptably high utility loss.

We now show how to construct black-box computational unlearning (Definition 5) from differential privacy. There are two main ways to accomplish this: to use differential privacy directly or to aggregate the outputs of models in a differentially private way. The theorem below captures both of these cases.

**Theorem 15** (Differentially private computational unlearning). *Let $\mathcal{L}$ be a forgetting learning scheme that achieves black-box computational unlearning. Let* unlearn *simply output the original model (with fresh randomness for the differentially private mechanism). Then* learn *and* unlearn *satisfy the definition of black-box computational unlearning (Definition 5) if and only if $\delta \leq \mathrm{negl}\,(\lambda)$ and let $\epsilon \leq \ln\,(1 + \mathrm{negl}\,(\lambda))$.*

Unfortunately this approach also has the following undesirable result:

**Corollary 16.** *Let $\mathcal{L}$ be a forgetting learning scheme that achieves black-box computational unlearning, with* learn *implemented as described in Theorem 15. Then $M_u$ and $M_o$ are also computationally indistinguishable. This implies that the utility of $M_o$ is equivalent to the utility of $M_u$.*

*Remark* 17 (Black-box infeasibilty implies white-box infeasibility.). The security notion of white-box computational unlearning in Definition 4 is strictly stronger than the black-box computational unlearning of Definition 5. Thus, the an infeasibility result for black-box computational unlearning immediately implies an infeasibility result for white-box computational unlearning.

We note that most use cases of differentially private infer algorithms are designed to support a number of queries bounded by a constant. One possible interpretation of our result is that we assume an adversary is able to query the model some polynomial number of times.

We additionally stress that Theorem 15 and Corollary 16 only consider applying differential privacy to the infer algorithm of a learning scheme. Our result does not necessarily imply a utility collapse for a forgetting learning scheme that achieves computational unlearning with a differentially private learn algorithm.

# 6 CONCLUSION

In summary, we have proposed computational unlearning, a new framework for evaluating machine unlearning. Computational unlearning is satisfied by an unlearning method if the output of the unlearning method is indistinguishable from a mirror (control) model. We rigorously define indistinguishability in terms of a novel two-party cryptographic protocol which captures an adversary's ability to distinguish between two models. Computational unlearning provides both empirical and theoretical contributions to the field of unlearning by improves upon prior evaluation methods, such as membership inference attack (MIA) scores.

We empirically showed that several machine unlearning methods from literature (Foster et al., 2023; Graves et al., 2020; Chundawat et al., 2023; Zhang et al., 2024) do not achieve computational unlearning by presenting multiple algorithms that allow an adversary to distinguish between the model produced by an unlearning method and a control model.

We have identified several theoretical implications that naturally follow from our formal definition of computational unlearning. For example, all unlearning methods that meet our definition of computational unlearning must be randomized; there are no deterministic computational unlearning methods despite there being several deterministic unlearning methods proposed in prior work. We also proved that building computational machine unlearning using differential privacy techniques leads to utility collapse.

We believe there are several directions for future work. For example, relaxations of our computational unlearning framework — such as letting the challenger delete additional information beyond what is selected by the adversary — may be worth exploring. Additionally, we believe future work should consider how to apply unlearning methods to align generative models and explore how to incorporate notions of foundation models into computational unlearning.

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

# A    Taxonomy of Unlearning Methods

We categorize machine unlearning methods in one of three ways: as *heuristic unlearning*, *approximate unlearning*, or *exact unlearning* as applied to classification models.

**Heuristic unlearning.**    Unlike exact and approximate unlearning methods, *heuristic unlearning methods* do not have any formal guarantees. However, they are typically much less expensive than applying differential privacy or retraining the model (Foster et al., 2023; Golatkar et al., 2019; Chundawat et al., 2023; Kodge et al., 2023; Tarun et al., 2021). These rely on various heuristics that aim to minimize an unlearning "score" that that attempts to capture how well a machine learning model has forgotten. Membership inference attacks (MIA) (Shokri et al., 2017) are a popular scoring method used in the literature.

We now describe three heuristic unlearning methods: *bad teacher unlearning* (Chundawat et al., 2023), *amnesiac unlearning* (Graves et al., 2020), and *selective synaptic dampening (SSD)* (Foster et al., 2023). Each of these heuristic unlearning methods are evaluated on membership inference attack (MIA) scores; this is representative of many heuristic unlearning methods.

- *Bad teacher unlearning.* Bad teacher unlearning rests on the assumption that, after forgetting a data point, a model's behavior on that data point should be similar to that of a randomly initialized model. To forget $\mathcal{D}_f$ the model is "taught" to reflect the behavior of a randomly initialized model (i.e. a *bad teacher*).

- *Amnesiac unlearning.* Amnesiac unlearning tries to reverse the changes to the model incurred by training on $\mathcal{D}_f$ by keeping track of all batches containing elements from $\mathcal{D}_f$; gradient *ascent* is performed on these training batches at forget time. This attempts to "backtrack" towards a model that never had those gradient updates applied. We note that this approximates the approaches taken by many exact unlearning methods.

- *Selective synaptic dampening (SSD).* SSD measures the $\mathcal{D}_f$-related information in each neuron by using the Fisher information matrix (FIM). Neurons that contain lots of information about examples in $\mathcal{D}_f$ are "zeroed out" by scaling down their weights. One can think of SSD as a pruning algorithm where "branches" of the network are "removed" based on their "knowledge" of $\mathcal{D}_f$.

**Approximate unlearning.**    An *approximate machine unlearning method* attempts to output a model that is approximately equal to a model trained without the forget set with high probability. Approximate machine unlearning methods are typically based on the notions of *differential privacy* (Dwork & Roth, 2014) and *certified removal* (Guo et al., 2023).

**Differential privacy.**    Differential privacy (Dwork & Roth, 2014) bounds the difference between outputs of a randomized algorithm on similar data sets. In the context of machine learning, this can be implemented as either (1) producing model parameters that are similar to the model parameters produced by training on a similar dataset or (2) producing an inference that is similar to the inference produced by a model trained on a similar dataset.

**Certified removal.**    Certified removal draws inspiration from the aforementioned notion of differential privacy, extending a white box privacy guarantee to hold for a learning and unlearning method. Their aim is to bound the difference in model's produced by the unlearning method and the model's produced by the learning method without a particular data point in the training set. We refer the reader to Guo et al. for the formal definition (Guo et al., 2023). Zhang et al. (Zhang et al., 2024) extend certified removal to non-linear models with non-convex objectives via *certified deep unlearning (CDU)* in order apply a certified removal technique to deep neural networks. We evaluate CDU in §4.

**Exact unlearning.**    An *exact unlearning method* modifies the original model such that its outputs exactly match a model trained without the forget set. We are unaware of any exact unlearning method for neural networks that does not involve some degree of retraining. The most common approaches rely on saving checkpoints of model state at train time (Bourtoule et al., 2020; Ullah et al., 2021). Unlearning then consists of rewinding to a checkpoint that has not been influenced by the forget set

and then resuming training from that point without the forgotten data. This technique is essentially a time-space tradeoff; multiple checkpoints of the model must be saved out during training. The worst-case retraining cost may be equivalent to retraining the model from scratch (for example, if the forget set contains an element from the first batch). We do not study exact unlearning in this work.

# B    IMPLEMENTATION DETAILS

All models used the ResNet-18 (He et al., 2015a) architecture. The original and control models were trained using stochastic gradient descent with momentum and weight decay. The hyperparameters used are as follows:

- Number of epochs: 50
- Batch size: 512
- Learning rate: $10^{-2}$
- Weight decay: $5 \times 10^{-4}$

For SSD (Foster et al., 2023), we used a dampening constant of 1 and a selection weighting of 100. For all other methods (Chundawat et al., 2023; Graves et al., 2020; Zhang et al., 2024), we used the parameters specified in their original papers (with the exception of $\sigma$ for CDU (Zhang et al., 2024), which we varied in §4).

# C    PROOFS

*Proof of Theorem 7.* Let `learn` be defined as normal for $k$-NN models. Let `unlearn` be defined as deleting the specified $\mathcal{D}_f$ from the $k$-NN database. Observe that this produces the same database as `learn` on $\mathcal{D} \setminus \mathcal{D}_f$. Therefore, an adversary cannot distinguish between $M_u$ and $M_c$ with non-negligible advantage because they are exactly the same model.    □

## C.1    DETERMINISTIC COMPUTATIONAL UNLEARNING DOES NOT EXIST

*Proof of Theorem 12.* Suppose that a forgetting learning scheme is entropic. Therefore, `learn` $\left(\text{init}\left(1^\lambda\right), \mathcal{D}\right)$ is a randomized algorithm that samples some $h \in \mathcal{H}$ with minimum entropy greater than 1 bit. Let $\mathbb{P}(h)$ be the probability that `learn` samples a particular $h \in \mathcal{H}$ and let

$$p_{\max} = \max_{\forall h \in \mathcal{H}} \mathbb{P}(h)$$

Now suppose that the challenger uses a deterministic `unlearn` algorithm. Then the adversary can also run `unlearn` on $M_o$ and will win the game if $M_c \neq M_u$. Because the probability `learn` will output a particular model is bounded by $p_{\max}$, the probability that $M_c = M_u$ is also bounded by $p_{\max}$ and the probability $M_c \neq M_u$ is at least $1 - p_{\max}$. Because `unlearn` is a computational unlearning algorithm, we must have that $1 - p_{\max} < \frac{1}{2} + \text{negl}(\lambda)$. We can rearrange symbols to get that $\text{negl}(\lambda) > \frac{1}{2} - p_{\max}$. But we have a contradiction because $p_{\max}$ does not asymptotically approach $\frac{1}{2}$ as $\lambda$ approaches infinity.    □

*Proof of 13.* Suppose that $\mathcal{L}$ is a deterministic learning scheme. Therefore, it must output a single model for a given training set $\mathcal{D}$. Suppose $\mathcal{L}$ is also forgetting and achieves computational unlearning. We now consider two possible cases: that `unlearn` is randomized and that it is deterministic.

- *Randomized case:* Suppose that `unlearn` is a randomized algorithm that samples some $h \in \mathcal{H}$. Let $\mathbb{P}(h)$ be the probability that `unlearn` selects a particular $h \in \mathcal{H}$ and let

$$p_{\max} = \max_{\forall h \in \mathcal{H}} \mathbb{P}(h)$$

Recall that in this scenario, the challenger uses a deterministic `learn` algorithm to produce $M_c$. Then the adversary can also run `learn` to produce $M_c$ and will win the game if

$M_c \neq M_u$. Because the probability `unlearn` will output a particular model is bounded by $p_{\max}$, the probability that $M_c = M_u$ is also bounded by $p_{\max}$ and the probability $M_c \neq M_u$ is at least $1 - p_{\max}$. Because `unlearn` is a computational unlearning algorithm, we must have that $1 - p_{\max} < \frac{1}{2} + \mathrm{negl}(\lambda)$. We can rearrange symbols to get that $\mathrm{negl}(\lambda) > \frac{1}{2} - p_{\max}$. But we have a contradiction because $p_{\max}$ does not asymptotically approach $\frac{1}{2}$ as $\lambda$ approaches infinity.

- *Deterministic case:* Now suppose that `unlearn` is a deterministic algorithm. Then the adversary can also run `learn` and `unlearn` on $M_o$ and will win the game if $M_c \neq M_u$. Because `learn` and `unlearn` are deterministic and will each output a particular model for a given dataset, we must have that $M_c = M_u$. Thus, `unlearn` must be a *perfect* unlearning algorithm.

$\square$

## C.2 BLACK-BOX COMPUTATIONAL UNLEARNING FROM DIFFERENTIAL PRIVACY COLLAPSES UTILITY

We begin by recalling the definition of privacy loss and differential privacy.

**Definition 18** (Privacy Loss, (Dwork & Roth, 2014)). The privacy loss $\mathcal{L}$ over neighboring databases $x, y$ after observing $\xi$ is given by:

$$\mathcal{L}^{(\xi)}_{\mathcal{M}(x) \| \mathcal{M}(y)} = \ln\left(\frac{\mathbb{P}(\mathcal{M}(x) = \xi)}{\mathbb{P}(\mathcal{M}(y) = \xi)}\right)$$

**Definition 19** (Differential Privacy, (Dwork & Roth, 2014)). A randomized algorithm $\mathcal{M}$ with domain $\mathbb{N}^{|\mathcal{X}|}$ is $(\epsilon, \delta)$-differentially private if for all $\mathcal{S} \subseteq \mathrm{Range}(\mathcal{M})$ and for all $x, y \in \mathbb{N}^{|\mathcal{X}|}$ such that $\|x - y\|_1 \leq 1$:

$$\mathbb{P}(\mathcal{M}(x) \in \mathcal{S}) \leq e^\epsilon \cdot \mathbb{P}(\mathcal{M}(y) \in \mathcal{S}) + \delta$$

If $\delta = 0$, we say that $\mathcal{M}$ is $\epsilon$-differentially private.

Differential privacy's definition bounds the privacy loss from any query, which we discuss below.

*Remark* 20 (Privacy Loss Bounded for Differentially Private Algorithms, (Dwork & Roth, 2014)). Suppose that $\mathcal{M}$ is a $(\epsilon, \delta)$-differentially private algorithm. Then by definition, the absolute value of the privacy loss $\mathcal{L}^{(\xi)}_{\mathcal{M}(x) \| \mathcal{M}(y)}$ is bounded by $\epsilon$ with probability at least $1 - \delta$.

*Remark* 21 (Differential Privacy is Immune to Post-Processing, (Dwork & Roth, 2014)). Additionally, one of the most useful properties of differential privacy is that it is "immune" to post-processing. This means that there exists no algorithm that, given the output of a differentially-private function, can "undo" the differential privacy. We refer the reader to (Dwork & Roth, 2014, Proposition 2.1) for the proof of this claim.

We will use this property to show that differential privacy can be used to satisfy the definition of black-box computational unlearning (Definition 5).

**Lemma 22.** *Privacy Loss is an upper bound on relative entropy.*

*Proof of Lemma 22.* Recall the definition of relative entropy (Kullback-Leibler divergence) of probability distribution $Q$ with respect to $P$ (Kullback & Leibler, 1951):

$$D_{\mathrm{KL}}(P \| Q) = \sum_{x \in \mathcal{X}} P(x) \ln\left(\frac{P(x)}{Q(x)}\right) \tag{2}$$

Now, suppose we have some randomized algorithm $\mathcal{M}$ with inputs $a, b$. Let $P, Q$ represent the output distributions of $\mathcal{M}(a), \mathcal{M}(b)$ respectively. Let $\mathcal{L}_{\max}$ refer to the maximum privacy loss observed for any element $x$.

$$(2) = \sum_{x \in \mathcal{X}} P(x) \ln \left( \frac{\mathbb{P}(\mathcal{M}(a) = x)}{\mathbb{P}(\mathcal{M}(b) = x)} \right)$$

$$= \sum_{x \in \mathcal{X}} P(x) \mathcal{L}_{\mathcal{M}(a) \parallel \mathcal{M}(b)}^{(x)}$$

$$\leq \sum_{x \in \mathcal{X}} \mathcal{L}_{\max}$$

Because $P$ is a probability distribution, we have that $P(x) \in [0, 1]$. Then privacy loss is an upper bound because the relative entropy is equal to the privacy loss multiplied by $P(x)$ by definition. □

*Proof of Theorem 15.* Observe that the privacy loss is negligible in $\lambda$ with overwhelming probability. This means that the relative entropy between the outputs of $M_u$ and $M_c$ is negligible by Lemma 22. By Remark 21, there is no algorithm an adversary can use to increase the relative entropy. So then $M_u$ and $M_c$ are computationally indistinguishable.

We now show that our bounds are tight. Suppose that $\delta > \text{negl}(\lambda)$. Then the privacy loss guarantee does not hold with overwhelming probability and an adversary could obtain a query result with non-negligible privacy loss after a polynomial number of queries.

Alternatively, suppose that $\epsilon > \ln(1 + \text{negl}(\lambda))$. Then the privacy loss guarantee is at least polynomial in $\lambda$ and an adversary could obtain query results that lead to a non-negligible privacy loss after a polynomial number of queries. □

*Proof of Corollary 16.* We follow the proof of Theorem 15. Observe that the privacy loss is negligible in $\lambda$ with overwhelming probability. This means that the relative entropy between the outputs of $M_u$ and $M_c$ is negligible. But $M_u$ is the same model as $M_o$, with fresh randomness for the differential privacy mechanism. So $M_u$ and $M_o$ are also computationally indistinguishable.

In other words, this means that $\texttt{util}(M_o) \overset{c}{\approx} \texttt{util}(M_u)$. Since $C$ does not know *a priori* the choice of $\mathcal{A}$, $\texttt{unlearn}$ must be indistinguishable for all possible choices. So then $M_u \overset{c}{\approx} M_c$ for $\mathcal{D}_f = \mathcal{D}$. That is to say that $M_u \overset{c}{\approx} \texttt{learn}\left(\texttt{init}\left(1^\lambda\right), \emptyset\right)$. But we because $\texttt{util}(M_o) \overset{c}{\approx} \texttt{util}(M_u)$ we also have $\texttt{util}(M_o) \overset{c}{\approx} \texttt{util}\left(\texttt{learn}\left(\texttt{init}\left(1^\lambda\right), \emptyset\right)\right)$, which is bounded by a small $\epsilon$ and thus not meaningful. □

