# OpenReview forum: "Mirror Mirror on the Wall, Have I Forgotten it All? A New Framework for Evaluating Machine Unlearning"
_ICLR.cc/2026/Conference — Submitted to ICLR 2026_

### Official Review · Reviewer_mPjn · 2025-10-27

**Soundness:** 1
**Presentation:** 1
**Contribution:** 1
**Rating:** 2
**Confidence:** 2

**Summary:**

This paper introduces a framework for evaluating machine unlearning called computational unlearning. The core intuition is that a truly unlearned model should be computationally indistinguishable from a model that was trained from scratch without the data to be forgotten (referred to as a "mirror" or "control" model). The paper includes both a theoretical analysis that proves the correctness and properties of computational unlearning, as well as experimental evaluation that tests whether existing unlearning methods in the literature satisfy the computational unlearning criterion.  In practice, it might be difficult to achieve computational unlearning. The paper would benefit from more detailed discussion of the specific obstacles that make computational unlearning challenging to achieve, the trade-offs between unlearning guarantees and model utility, and potential pathways toward developing more effective unlearning algorithms that can satisfy computational indistinguishability while remaining computationally feasible for large-scale machine learning systems.

**Strengths:**

1. Machine unlearning is an interesting topic and leaves many open questions. The authors propose a new machine unlearning terminology.

2. The proposed computational unlearning is formulated as a security game between a Challenger and an Adversary.

3. The authors provide a theoretical analysis.

**Weaknesses:**

1. The connection between computational indistinguishability and machine unlearning verification remains insufficiently justified in the current presentation. While computational indistinguishability is a well-established cryptographic concept that measures whether two probability distributions can be efficiently distinguished by any polynomial-time algorithm, the paper does not adequately explain how this framework translates to proving that data has been successfully unlearned from a model. The authors need to provide a more rigorous theoretical foundation that bridges the gap between these two concepts, perhaps by explicitly showing how an adversary's inability to distinguish between an unlearned model and a retrained model formally guarantees that no information about the forgotten data remains accessible.

2. Section 2.2 "Motivation with k-NN" lacks clarity and fails to provide intuitive reasoning for why the k-nearest neighbors algorithm serves as an appropriate motivating example for computational indistinguishability in the unlearning context. The authors appear to assume that the relevance of k-NN to their framework is self-evident, but the connection is actually quite subtle and requires careful exposition. The authors can consider adding concrete examples or toy problems that demonstrate why k-NN specifically illuminates the core challenges of indistinguishability-based unlearning, rather than simply presenting it as a given starting point for their theoretical development.

3. A significant weakness of the paper is its failure to address the practical applications and real-world implications of computational unlearning. While the theoretical framework may be sound, how this approach would be implemented in actual machine learning systems and what benefits it would provide to practitioners is unclear. The authors should dedicate space to discussing concrete use cases, potential computational costs, scalability considerations, and how their method compares to existing unlearning techniques in terms of both effectiveness and efficiency. Additionally, they should address practical concerns such as how frequently unlearning requests might occur in real systems, whether the method can handle multiple simultaneous unlearning requests, and what guarantees can be provided to users seeking to have their data removed from trained models.

**Questions:**

1. Can you elaborate on why to use kNN to motivate the indistinguishability?
2. Do you have any practical suggestions for improving existing machine unlearning methods?

---

### Official Review · Reviewer_WBha · 2025-10-29

**Soundness:** 2
**Presentation:** 2
**Contribution:** 2
**Rating:** 2
**Confidence:** 3

**Summary:**

This paper proposes a new framework of computational unlearning, targeting the evaluation of machine unlearning methods. It proposes that when an efficient Adversary is unable to distinguish between the unlearned model and the control model, the unlearning method achieves computational unlearning. Empirically, the paper shows that many unlearning methods fail this standard. Theoretically, the framework demonstrates that no deterministic unlearning algorithms exist for common entropic learning schemes, and that achieving computational unlearning via Differential Privacy leads to a collapse in model utility.

**Strengths:**

1. Propose a framework based on cryptographic theory for evaluating unlearning methods, which rigorously define the criteria for successful unlearning as computational indistinguishability between the unlearned model and a control model.

2. The paper makes strong theoretical contributions by leveraging a security game framework rooted in cryptography to rigorously define and evaluate machine unlearning. The conclusion that none of the deterministic, heuristic unlearning methods can achieve computational unlearning for entropic learning schemes, and the proof of achieving computational unlearning using strong differential privacy method leads to utility collapse are very interesting.

**Weaknesses:**

1. The computational unlearning evaluation based only serves to assess the efficacy of unlearning by measuring posterior difference (MIAScore and KLDScore) from the ideal retrained model, without offering constructive insights or a method for developing better approximate unlearning solutions.

2. The paper lacks a dedicated comparison or discussion of its framework against prior machine unlearning evaluation works that similarly employ game-theoretic or cryptographic-inspired methodologies[1].

3. While the paper formally defines the black-box computational unlearning security game, it contains no experiments to empirically validate the effectiveness of its proposed distinguishing algorithms (MIAScore and KLDScore) in this crucial real-world scenario.

4. The distinguisher scores is a important component in the proposed framework, because the performance of the proposed framework rely heavily on the employed scores. However, both of the MIAScore and KLDivergence score have been extensively employed in prior machine unelarning research (e.g. SSD and Bad Teacher), which limit the novelty of this paper.

5. Although the paper provides a rigorous theoretical analysis, a practical weakness is that it fails to offer an explicit metric or clear workflow for comparison and improvement among unlearning methods. Instead, the authors only empirically demonstrate that the deployed methods fail to achieve the standard of computational unlearning (in Figure 2), without explicitly clarifying which method performs better.

References:

[1]  A Reliable Cryptographic Framework for Empirical Machine Unlearning Evaluation. NeurIPS 2025.

**Questions:**

1. Could you suggest potential improvements or solutions to enhance existing unlearning methods based on the given evaluation framework?

2. See weakness 2.

3. Since the performance of the proposed framework depends heavily on the distinguisher scores, how can we determine whether to use MIAScore or KLDScore in real-world scenarios?

4. In Figure 2, there is an abnormal drop in the success rate of SSD and CDU between unlearning sizes of 10 and 100. It is very interesting. Could you explain the underlying reason behind this behavior?

---

### Official Review · Reviewer_nFac · 2025-10-31

**Soundness:** 2
**Presentation:** 2
**Contribution:** 2
**Rating:** 2
**Confidence:** 3

**Summary:**

This paper introduces a new framework for evaluating machine unlearning. The paper proposes the formalization of computational unlearning, a strengthened notion of unlearning grounded in the computational indistinguishability between the outputs of an unlearned model and those of a model retrained from scratch. The framework employs measures based on the similarity between the two model outputs and supports empirical validation. The paper also discusses several theoretical implications arising from the proposed evaluation paradigm.

**Strengths:**

1. The development of rigorous evaluation methodologies for machine unlearning is a critically important research direction.
2.  The framework leverages two distinguisher scores, including a Membership Inference Attack (MIA)-based metric and a Kullback-Leibler (KL) divergence-based metric.

**Weaknesses:**

1. The empirical validation seems to be limited to a single model (ResNet-18) and dataset (CIFAR-10), which restricts the generalizability of the claims.
2. The presentation, particularly in the methodology section, lacks clarity and is difficult to follow.
3. Retraining from scratch inherently involves stochasticity. The methodology appears not to explicitly account for the randomness in the training dynamics.
4. The employed Membership Inference Attack can be strengthened. State-of-the-art approaches, such as LiRA-based attacks, would provide a more accurate and reliable evaluation.
5. Although the overforgetting and underforgetting are discussed in the motivation, the paper does not provide corresponding illustrations using the proposed evaluation framework.

**Questions:**

1. In Definition 5, how are the oracle models $O_{M_c}$ and $O_{M_u}$ formally instantiated?
2. For the KL-based score, what is the rationale behind injecting Gaussian noise with mean 0 and variance 0.1?

---

### Official Review · Reviewer_fRVR · 2025-11-04

**Soundness:** 2
**Presentation:** 2
**Contribution:** 3
**Rating:** 2
**Confidence:** 2

**Summary:**

This paper addresses an interesting problem: an adversary may still be able to distinguish between the unlearned model and the control model after the unlearning process. To better evaluate unlearning performance, the paper proposes a framework based on evaluation scores. The authors demonstrate the effectiveness of the proposed score on the CIFAR-10 dataset using several unlearning methods.

**Strengths:**

1, The authors present a very interesting perspective on machine unlearning evaluation.
2. It is commendable that the authors incorporate knowledge from other domains into the machine learning context, as mentioned in Line 187.

**Weaknesses:**

1. The presentation of this paper is difficult to follow. For example, it would be clearer to present the content in Figure 1 using an algorithmic block or pseudocode format.
2. The baseline methods selected by the authors are not sufficiently representative. Several well-known unlearning methods, such as SalUN [1] and $\ell_1$-sparse [2], could be included for a more comprehensive comparison.
3. It would be helpful to include a notation table in the Appendix, as many symbols and variables are used throughout the paper.

[1] Fan, Chongyu, et al. "Salun: Empowering machine unlearning via gradient-based weight saliency in both image classification and generation." arXiv preprint arXiv:2310.12508 (2023).

[2] Jia, Jinghan, et al. "Model sparsity can simplify machine unlearning." Advances in Neural Information Processing Systems 36 (2023): 51584-51605.

**Questions:**

1. What is the meaning of $C$ in definition 4?
2. What is the meaning of control model? Is it the same as the retrained model?

---

### Meta-Review · Area_Chair_ni6D · 2026-01-07

**Summary:**

The paper introduces "Computational Unlearning," a formal evaluation framework: The core premise is that a successful unlearning algorithm should produce a model that an adversary cannot distinguish from a mirror (control) model retrained from scratch. The authors contribute a theoretical analysis suggesting that deterministic unlearning is impossible for entropic learning schemes and that achieving this definition via Differential Privacy leads to utility collapse. Empirically, they demonstrate that several existing heuristic unlearning methods fail this indistinguishability test on ResNet-18 models trained on CIFAR-10, using Membership Inference Attack (MIA) and KL-Divergence scores as distinguishers.

**Reviewer Concerns:**

no response, all reviewers' concerns are there.

1. Limited Empirical Validation: Multiple reviewers [Reviewer nFac, Reviewer fRVR] criticized the experimental scope, noting that results are limited to a single architecture (ResNet-18) and dataset (CIFAR-10), which severely limits generalizability.

2. Clarity and Presentation: Significant confusion remained regarding core definitions. Reviewer mPjn found the k-NN motivation opaque; Reviewer fRVR requested clarification on the "control model" definition; Reviewer nFac found the methodology section difficult to follow.

3. Handling of Stochasticity: Reviewer nFac raised a critical point regarding how the framework accounts for the inherent randomness (stochasticity) of retraining from scratch, which complicates the "indistinguishability" claims if not rigorously modeled.

4. Missing Baselines: Reviewer fRVR noted the absence of relevant baselines such as SalUN and sparse unlearning methods.

**Reviewer Scores:**

no changes as no responses

---

### Decision · Program_Chairs · 2026-01-26

Reject